# ENHANCING DATA EFFICIENCY IN REINFORCEMENT LEARNING: A NOVEL IMAGINATION MECHANISM BASED ON MESH INFORMATION PROPAGATION

## ABSTRACT

Reinforcement learning(RL) algorithms face the challenge of limited data efficiency, particularly when dealing with high-dimensional state spaces and large-scale problems. Most of RL methods often rely solely on state transition information within the same episode when updating the agent's Critic, which can lead to low data efficiency and sub-optimal training time consumption. Inspired by human-like analogical reasoning abilities, we introduce a novel mesh information propagation mechanism, termed the 'Imagination Mechanism (IM)', designed to significantly enhance the data efficiency of RL algorithms. Specifically, IM enables information generated by a single sample to be effectively broadcasted to different states across episodes, instead of simply transmitting in the same episode. This capability enhances the model's comprehension of state interdependencies and facilitates more efficient learning of limited sample information. To promote versatility, we extend the IM to function as a plug-and-play module that can be seamlessly and fluidly integrated into other widely adopted RL algorithms. Our experiments demonstrate that IM consistently boosts four mainstream SOTA RL algorithms, such as SAC, PPO, DDPG, and DQN, by a considerable margin, ultimately leading to superior performance than before across various tasks. For access to our code and data, please visit https://github.com/OuAzusaKou/IM.

## 1 INTRODUCTION

Data efficiency has been a fundamental and long-standing problem in the field of reinforcement learning (RL), especially when dealing with high-dimensional state spaces and large-scale problems. While RL has shown great promise in solving complex problems, it often requires a large amount of data, making it impractical or costly in many real-world applications. The sample complexity of such state-of-the-art agents is often incredibly high: MuZero (Schrittwieser et al., 2020) and Agent-57 (Badia et al., 2020) use 10-50 years of experience per Atari game, and (Berner et al., 2019) uses 45,000 years of experience to accomplish its remarkable performance. This is clearly impractical: unlike easily-simulated environments such as video games, collecting interaction data for many real-world tasks is extremely expensive. Therefore, improving data efficiency is crucial for RL algorithms (Dulac-Arnold et al., 2019).

To deal with data efficiency challenge. RAD (Laskin et al., 2020b)introduces two new data augmentation methods: random translation for image-based input and random amplitude scaling for proprioceptive input. CURL (Laskin et al., 2020a) performs contrastive learning simultaneously with an off-policy RL algorithm to improve data efficiency over prior pixel-based methods. (Schwarzer et al., 2021) addresses the challenge of data efficiency in deep RL by proposing a method that uses unlabeled data to pretrain an encoder, which is then finetuned on a small amount of task-specific data.

While these achievements are truly impressive, it's important to note that current RL algorithms acquire information from state transition samples through interactions with the environment and then this information still can only be propagated and utilized within the same episode(as shown in Figure.1), by Temporal Difference (TD) updates (Sutton & Barto, 2018). This may result in the inability to propagate and utilize information contained in states from different episodes, thereby lowering the data efficiency of RL algorithms.

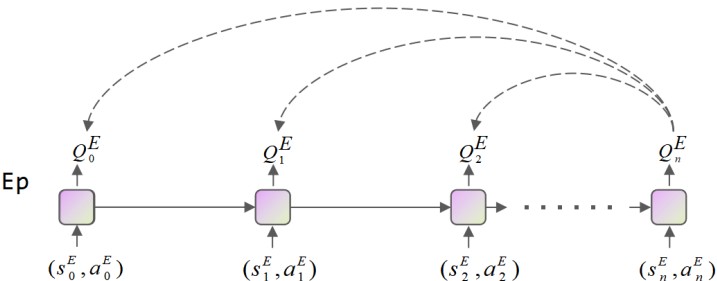

Figure 1: Information propagation path in the TD updates. It can be observed that information contained in $Q_n^E$ for a specific state-action pair $(s_n^E, a_n^E)$ can only propagate to other $Q_i^E$ within the same episode through TD updates. $Q_i^E$ represent the Critic value of state-action pair $(s_i^E, a_i^E)$. Each episode $Ep$ comprises a sequence of states $s_i^E$, actions $a_i^E$, at each time step $i$, $i \in (0, 1, 2, \ldots, n)$.

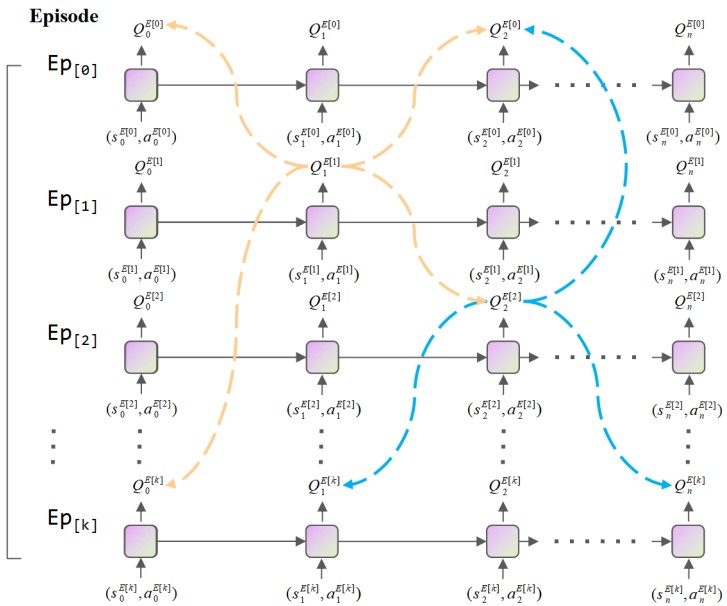

Figure 2: Information propagation path in the Imagination Mechanism. It's apparent that information contained in $Q_n^{E[k]}$ pertaining to a state-action pair $(s_n^{E[k]}, a_n^{E[k]})$ can be broadcasted to different $Q_i^{E[j]}$ across episodes through the utilization of the IM, where $i \in (0, 1, 2, \ldots, n)$ and $j \in (0, 1, 2, \ldots, k)$. As shown in the figure, for instance, instead of merely transmitting within the same episode, the information $Q_1^{E[1]}$ associated with $(s_1^{E[1]}, a_1^{E[1]})$ and $Q_2^{E[2]}$ associated with $(s_2^{E[2]}, a_2^{E[2]})$ can propagate to other Critic value $Q_i^{E[j]}$ across episodes. The propagation paths are represented by the yellow and blue dashed lines, respectively.

Human beings can leverage their experiences in one task to improve their performance in another task through analogical reasoning. Inspired by human-like analogical reasoning abilities (Leonard et al., 2023; Sternberg, 1977; Sternberg & Rifkin, 1979), we propose an IM that enables the mutual propagation of information contained in states across different episodes(as shown in Figure.2). Specifically, we introduce a similarity-based difference inference module, which infers the Critic difference between two states based on the similarity of them, where Critic represents the value function in RL. After updating the Critic value of one state, we employ this module to broadcast the Critic's change values to other states' Critic, thereby enhancing the estimation efficiency of the value function.

To this end, we propose IM based on a Similarity Calculation Network and a Difference Inference Network, and we make it a general module that can be theoretically applied in almost any deep RL algorithms. Furthermore, we conduct extensive experiments to validate our concept. The contributions of this paper are summarized as follows:

- We propose IM, consisting of a Similarity Calculation Network and a Difference Inference Network. IM enables mutual information propagation across episodes, significantly enhancing data efficiency in various tasks. To the best of our knowledge, our mechanism has not been employed in prior work.

- To promote versatility, we extend IM to function as a plug-and-play module that can be seamlessly and fluidly integrated into other widely adopted RL methods.

- Extensive experiments show that IM consistently boosts four mainstream RL-algorithms, such as SAC, PPO, DDPG, and DQN, by a considerable margin, ultimately leading to superior performance(SOTA) than before in terms of data efficiency across various tested environments.

## 2 RELATED WORK

### 2.1 MAINSTREAM RL ALGORITHMS

DQN (Van Hasselt et al., 2016) is a classic RL algorithm based on value functions used to solve discrete control problems. It employs experience replay and target networks to stabilize training. DDPG (Silver et al., 2014) is a classic Actor-Critic architecture method designed for continuous action space. In contrast to the off-policy methods mentioned above, on-policy methods sacrifice sample efficiency but enhance training stability. Techniques like TRPO (Schulman et al., 2015) utilize trust regions to limit the size of policy updates and leverage importance sampling to improve training stability. PPO (Schulman et al., 2017) provides a more concise way to restrict policy updates and has shown better practical results. SAC (Haarnoja et al., 2018) is a relatively newer off-policy method that introduces an entropy regularization term to encourage policy exploration in uncharted territories. It also uses the minimum value from dual Q-networks to estimate Q-values, avoiding overestimation to enhance training stability. In our work, we use these four RL algorithms as baselines to validate the effectiveness of IM.

### 2.2 DATA EFFICIENCY

A significant amount of research has been conducted to enhance the data efficiency in RL. SiM-PLe (Kaiser et al., 2019) develops a pixel-level transition model for Atari games to generate simulated training data, achieving remarkable performance in the 100k frame setting. However, this approach demands several weeks of training. Variants of Rainbow (Hessel et al., 2018), namely DER (Van Hasselt et al., 2019) and OTRainbow (Kielak, 2019), are introduced with a focus on improving data efficiency. In the realm of continuous control, multiple studies (Hafner et al., 2019; Lee et al., 2020) suggest the utilization of a latent-space model trained with a reconstruction loss to boost data efficiency. Recently, in the field of RL, DrQ (Yarats et al., 2021) and RAD (Laskin et al., 2020b) observe that the application of mild image augmentation can significantly enhance data efficiency, yielding superior results to previous model-based approaches. CURL (Laskin et al., 2020a) proposed a combination of image augmentation and a contrastive loss to improve data efficiency for RL. SPR (Schwarzer et al., 2020) trains an agent to predict its own latent state representations multiple steps into the future. SGI (Schwarzer et al., 2021) improves data efficiency by using unlabeled data to pretrain an encoder which is then finetuned on a small amount of task-specific data.

Most of the work mentioned above primarily focuses on improving data efficiency in pixel-based RL, which has certain limitations and is not a universally applicable method for enhancing data efficiency. Additionally, the methods mentioned earlier still rely on TD updates, we argue that using TD updates alone to update the Critic can result in inadequate information utilization, even leading to catastrophic learning failures, as previously discussed. To address these issues, we propose a general mesh information propagation mechanism, termed as the 'Imagination Mechanism (IM),' designed to significantly enhance the data efficiency of RL algorithms.

## 3  METHOD

Traditional RL algorithms update the Critic using the TD updates. We contend that TD updates can only propagate state transition information to previous states within the same episode, thereby limiting the data efficiency of RL algorithms, as demonstrated in Figure.1. It is well known that human beings can utilize their experiences from one task to another through analogical reasoning. Therefore, motivated by human-like analogical reasoning abilities, we propose IM that employs a sequential process of comparison followed by inference. Specifically, (1) we utilize a similarity calculation network(SCN) to compare states, yielding their respective similarity scores. (2) We design a difference inference network(DIN) to perform inference based on the outcomes of the comparison(similarity scores). (3) Finally, leveraging a mesh structure for information propagation, we can transmit information from a state transition sample to any state across episodes to update the Critic, as presented in Figure.2. The following subsections will cover SCN, DIN, and the application of IM in other RL algorithms, respectively.

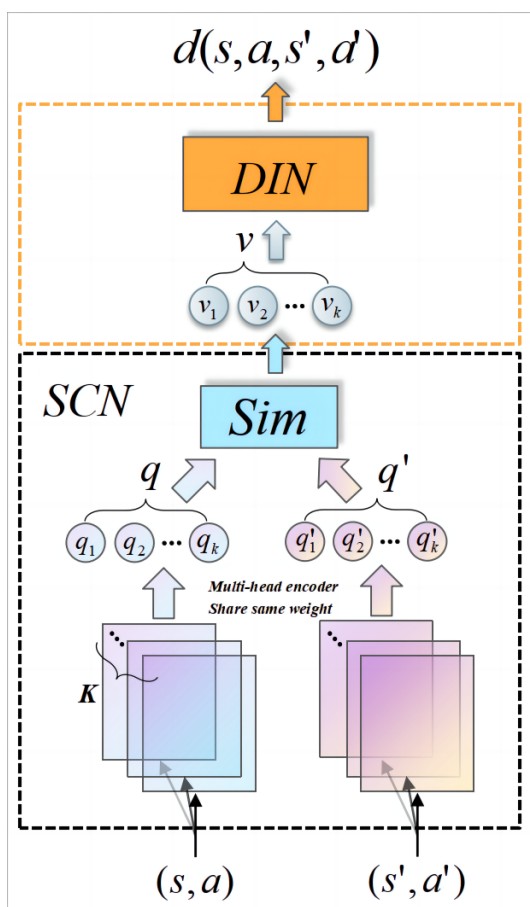

Figure 3: Overview of the IM Framework. IM comprises the Similarity Calculation Network (SCN) and the Difference Inference Network (DIN). For a detailed workflow, please refer to the following sections.

### 3.1  SIMILARITY CALCULATION NETWORK

The intention of the SCN is to calculate the similarity feature. As shown in the dashed black rectangle in Figure.3. For any two given pairs $(s, a), (s^{'}, a^{'})$, they are processed through multi-head encoder $f_i$, to extract features $q$ and $q^{'}$, where $i \in 1, 2, \ldots, k$. Subsequently, we calculate the similarity between $q$ and $q^{'}$ using function $Sim$ to obtain similarity vector $v$. The $Sim$ function can be employed with

various similarity methods, such as bi-linear inner-product or cosine similarity. The specific process is as follows:

$$q_i = f_i(s, a) \ , \ q_i' \ = f_i(s', a') \tag{1}$$

$$v_i = Sim(q_i, q_i') \tag{2}$$

In our work, we use cosine similarity as "$Sim$".

## 3.2 DIFFERENCE INFERENCE NETWORK

The purpose of the Difference Inference Network(DIN) is to infer difference $d$ between the Critics of two state-action pairs $(s, a)$ and $(s', a')$ based on similarity vector $v$. As shown in the dashed orange rectangle in Figure.3. Specifically, DIN takes $v$ as input and employs MLP to calculate the difference $d$ between the Critics for the two different state-action pairs. Finally, we utilize $Q(s, a)$ along with $d(s, a, s', a')$ to infer $Q(s', a')$. The specific process is as follows:

$$d(s, a, s', a') = MLP(v) \ , \ v = [v_1, v_2, \dots, v_k] \tag{3}$$

$$Q(s', a') \leftarrow Q(s, a) + d(s, a, s', a') \tag{4}$$

## 3.3 THE PROCESS OF APPLYING IM IN RL ALGORITHMS

Firstly, RL algorithm interacts with the environment to collect sample data. Next, these sample data are used to update the Critic or Actor. Afterward, we employ IM to update Critic. Finally, the training process is accomplished through iterations of the aforementioned steps. Below is a pseudocode example using the SAC algorithm.

---

**Algorithm 1** Soft Actor-Critic with Imagination Mechanism

---
    Initialize actor, critic and replay buffer.
    **for** each iteration **do**
        **for** each environment step **do**
            $\mathbf{a}_t \sim \pi_\phi(\mathbf{a}_t | \mathbf{s}_t)$
            $\mathbf{s}_{t+1} \sim p(\mathbf{s}_{t+1} | \mathbf{s}_t, \mathbf{a}_t)$
            $\mathcal{D} \leftarrow \mathcal{D} \cup \{(\mathbf{s}_t, \mathbf{a}_t, r(\mathbf{s}_t, \mathbf{a}_t), \mathbf{s}_{t+1})\}$
        **end for**
        **for** each gradient step **do**
            *update Actor and Critic()*
            *update Critic by IM()*
        **end for**
    **end for**

---

**IM pseudocode process:** (1) Given the input $(s, a)$, we randomly sample from the replaybuffer to obtain $(s_n, a_n)$. (2) Next, we feed both $(s, a)$ and $(s_n, a_n)$ into the feature encoder to obtain features $(q, q_n)$. (3) Then, we use the similarity function $Sim$ to obtain similarity vector $v$. (4)Similarity vector $v$ is passed to the DIN for Critic difference inference, resulting in the difference value $d$. (5) Finally, we use the Mean Squared Error($MSE$) between '$Critic(s, a) + d$' and '$Critic(s', a')$' as the loss function. The $Adam$ optimizer (He et al., 2020) is employed to update the parameters of $f_c$, $FC$, and the Critic, while the $momentum - averaged$ optimizer (Kingma & Ba, 2014) is used for updating the parameters of $f_d$.

```
1  # s,a: cuurent state and action
2  # s_n,a_n: other state and action
3  # f_c, f_d: feature encoder networks
```

```
4   # Sim: similarity method
5   # FC: full-connected layer
6   # loader: minibatch sampler from ReplayBuffer
7   # m: momentum, e.g. 0.95
8   # k: head num for Sim
9   # feature_dim: feature dimension
10  # q,q_n: shape: [B,k*feature_dim]
11  # Critic: State-Action function
12
13  f_c.params = f_d.params
14
15  for s_n,a_n in loader: # load minibatch from replay buffer
16
17      q = f_c.forward(s,a)
18      q_n = f_d.forward(s_n,a_n)
19      q_n = q_n.detach() # stop gradient
20      for i in range(k):
21
22          v[i] = Sim(q[i*feature_dim:(i+1)*feature_dim], q_n[i*
                  feature_dim:(i+1)*feature_dim])
23
24      d = FC(v)
25
26      loss = MSE(Critic(s,a) + d, Critic(s_n,a_n))
27      loss.backward()
28      update(f_c.params) # Adam
29      update(FC.params) # Adam
30      update(Critic.params) # Adam
31      f_d.params = m*f_d.params+(1-m)*f_c.params # momentum averaged
```

Listing 1: IM Learning Pseudocode(Pytorch-like)

## 4 EXPERIMENTS

### 4.1 EVALUATION

We measure our proposed method on five different tasks for both data efficiency and performance, using two different environment step sizes: 100k and 500k steps. We use a 500k step size because most environments reach asymptotic performance at this point. The 100k step size is used to assess the initial learning speed of the algorithms.

We evaluate (i) sample efficiency by measuring the number of steps it takes for the best-performing baselines to reach the same performance level as baselines+IM within a fixed T (100k) steps (as illustrated in Figure.5), and (ii) performance by calculating the ratio of episode returns achieved by baselines+IM compared to the vanilla baseline at T steps(as presented in Table 1).

### 4.2 ENVIRONMENTS

The primary objective of IM is to facilitate RL methods to be more data-efficient and effective, with broad applicability across a range of environments. We conduct evaluations using both discrete and continuous environments. Specifically, the continuous environments include "Ant", "Half Cheetah", and "Pendulum" from Mujoco (Todorov et al., 2012), while the discrete environments include "Lunar Lander" and "Acrobot" (Sutton, 1995) from the Gym (Brockman et al., 2016).

**Continuous environment :** Existing model-free RL algorithms often exhibit poor data efficiency when applied to Mujoco tasks, primarily due to the high-dimensional state spaces. Consequently, we embed IM into these baseline methods in three continuous control tasks within Mujoco: Half Cheetah, Ant, and Pendulum, with the aim of improving the data efficiency of these baselines.

Table 1: Scores(episode returns) achieved by IM combined with baselines and baselines on five tasks at environment steps T(100k and 500k). Our baselines are DQN, DDPG, PPO, and SAC. We also run IM with 10 random seeds given that these benchmark is susceptible to high variance across multiple runs.

| 100K STEP SCORES | Half Cheetah-V3 | Ant-V3 | Pendulum-V0 | Acrobot-V1 | Lunar Lander-V2 |
|---|---|---|---|---|---|
| SAC | 5228 ± 70 | 871 ± 150 | -249 ± 80 | - | - |
| SAC+Ours | 6652 ± 90 | 1627 ± 160 | -246 ± 87 | - | - |
| **Promotion** | **27.24%** | **86.79%** | **1.22%** | - | - |
| PPO | 206 ± 62 | 23 ± 5 | -235 ± 59 | - | - |
| PPO+Ours | 273 ± 52 | 37 ± 7 | -232 ± 73 | - | - |
| **Promotion** | **24.55%** | **60.87%** | **1.29%** | - | - |
| DDPG | 2523 ± 112 | -126 ± 12 | -250 ± 21 | - | - |
| DDPG+Ours | 3620 ± 96 | 16.38 ± 5 | -242 ± 16 | - | - |
| **Promotion** | **30.30%** | **87.00%** | **3.20%** | - | - |
| DQN | - | - | - | -86.60 ± 12 | 280 ± 13 |
| DQN+Ours | - | - | - | -83.12 ± 5 | 286 ± 7 |
| **Promotion** | - | - | - | **4.01%** | **2.14%** |
| 500K STEP SCORES | Half Cheetah-V3 | Ant-V3 | Pendulum-V0 | Acrobot-V1 | Lunar Lander-V2 |
| SAC | 10250 ± 242 | 2662 ± 137 | -246 ± 5 | - | - |
| SAC+Ours | 11877 ± 112 | 3654 ± 92 | -243 ± 2 | - | - |
| **Promotion** | **15.88%** | **32.27%** | **1.22%** | - | - |
| PPO | 1206 ± 35 | 231 ± 36 | -232 ± 3 | - | - |
| PPO+Ours | 1358 ± 22 | 302 ± 74 | -230 ± 1 | - | - |
| **Promotion** | **12.61%** | **30.74%** | **0.86%** | - | - |
| DDPG | 8831 ± 235 | 457 ± 267 | -247 ± 4 | - | - |
| DDPG+Ours | 10366 ± 272 | 562 ± 162 | -242 ± 2 | - | - |
| **Promotion** | **17.39%** | **22.98%** | **2.02%** | - | - |
| DQN | - | - | - | -83.94 ± 6 | 280 ± 9 |
| DQN+Ours | - | - | - | -82.95 ± 5 | 287 ± 7 |
| **Promotion** | - | - | - | **1.17%** | **2.50%** |

**Discrete environment :** We evaluate DQN in discrete control tasks, such as Acrobot and Lunar Lander, to ensure that our approach maintains generality in discrete control tasks as well.

## 4.3 BASELINES FOR BENCHMARKING DATA EFFICIENCY

As shown in the Table.1, we conduct comparisons with a total of four baseline methods, namely SAC, PPO, DDPG, and DQN, to validate IM in improving both the data efficiency and performance of RL algorithms. DQN is a classic RL algorithm based on value functions used to solve discrete control problems. It employs experience replay and target networks to stabilize training. DDPG is a classic Actor-Critic architecture method designed for continuous action spaces. In contrast to the off-policy methods mentioned above, on-policy methods sacrifice sample efficiency but enhance training stability. Techniques like TRPO utilize trust regions to restrict the size of policy updates and leverage importance sampling to improve training stability. PPO provides a more concise way to restrict policy updates and has shown better practical results. SAC is a relatively newer off-policy method that introduces an entropy regularization term to encourage policy exploration in uncharted territories. It also uses the minimum value from dual Q-networks to estimate Q-values, avoiding overestimation to enhance training stability.

## 4.4 RESULTS

Table.1 demonstrates the performance of four mainstream RL algorithms before and after the integration of IM at fixed steps T (100k or 500k). Specifically, in comparison to vanilla baselines, in three continuous tasks at the 100k setting, SAC+IM exhibits improvements of (27.24%, 86.79%, 1.22%), PPO+IM shows similar improvements of (24.55%, 60.87%, 1.29%), and DDPG+IM demonstrates parallel improvements of (30.30%, 87.00%, 3.26%). At the 500k setting, SAC+IM achieves enhancements of (15.88%, 37.27%, 1.22%) across all three environments, DDPG+IM presents improvements of (12.61%, 30.74%, 0.86%), and PPO+IM demonstrates parallel improvements of (17.39%, 22.98%, 2.02%). In two discrete tasks at the 100k setting, DQN+IM displays enhancements of (4.01%, 2.14%), and at the 500k setting, DQN+IM shows improvements of (1.17%, 2.50%). The results demonstrate that our approach yields conspicuous improvements across a spectrum of tasks. The significant improvement observed in tasks Ant and Half-Cheetah can be attributed to its

high-dimensional state space. It is evident that IM proves particularly effective in tackling challenges associated with such large-scale state spaces. In contrast, in environments such as Pendulum and Arcobot, characterized by relatively modest dimensional state spaces, the integration of IM does not yield substantial improvements. In scenarios such as Pendulum and Acrobot, vanilla RL algorithms have already reached high-performance levels, leaving little room for significant further improvement. A 500k step size is chosen since many environments reach their asymptotic performance at this point, while the 100k step size is employed to evaluate the initial learning speed of the algorithms (as mentioned in Sec 4.1).

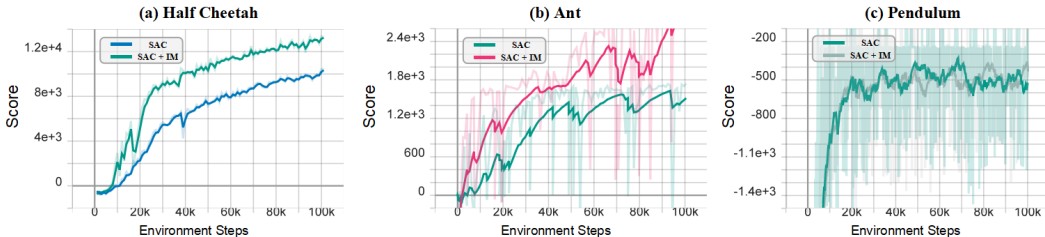

Figure 4: Training process of SAC and SAC+IM. It illustrates the changes in episode rewards concerning environment steps during training for IM incorporated into the SAC algorithm at the 100k step configuration across three tasks.

In complex, high-dimensional observation space scenarios, as mentioned earlier, the imagination mechanism can significantly enhance data efficiency and performance during training, as shown in Figure.4 (a) and Figure.4 (b).

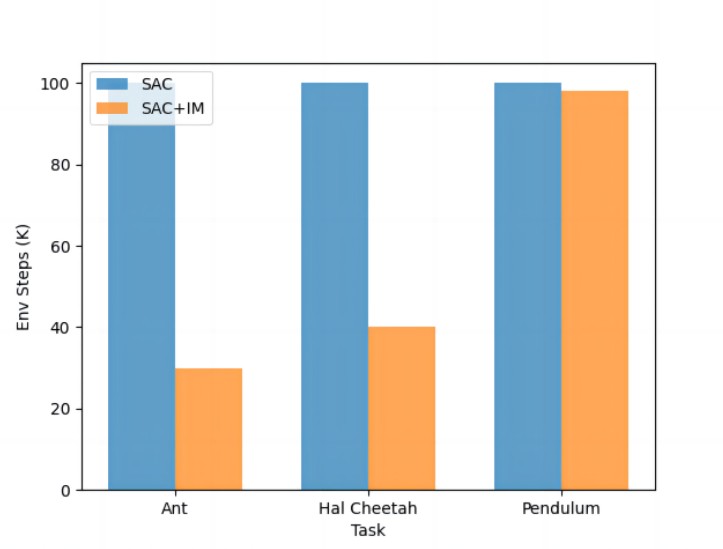

Figure 5: Data efficiency description. Environment steps SAC+IM requires to attain the performance level achieved by vanilla SAC at T=100k.

As illustrated in Figure 5, in the task 'Ant', SAC+IM achieves the performance of vanilla SAC in approximately 0.3x the number of T steps. Similarly, in the 'Hal cheetah' task, SAC+IM accomplishes this in roughly 0.4x the number of T steps. In the 'Pendulum' task, due to the modest dimensional state spaces, RL algorithms can reach high-performance levels, leaving little room for further improvement. Therefore, the impact of IM on such tasks is not obvious.

## 5 Conclusion

We contend that TD updates can only propagate state transition information to previous states within the same episode, thereby limiting data efficiency of RL algorithms. Inspired by human-like analogical reasoning abilities, we propose a simple plug-and-play module(IM) designed to significantly enhance the performance and data efficiency of any RL method. Our implementation is straightforward, plug-and-play, efficient, and has been open-sourced. We hope that IM's performance improvements, ease of implementation, and efficiency in real-world time utilization will become valuable assets for advancing research in data-efficient and generalizable RL methods.

## 6 Broader Impact

While significant progress has been made in fields such as Large Language Models (LLMs) (Floridi & Chiriatti, 2020), Computer Vision (CV) (He et al., 2016), Natural Language Processing (NLP) (Cambria & White, 2014), Reinforcement Learning (RL) (Silver et al., 2016), and Recommender Systems (RC) (Cambria & White, 2014), in terms of hardware (e.g., GPU acceleration from Nvidia) and software algorithms (e.g., ResNet, BERT, Transformer), there are pressing concerns associated with the increasing data costs and the environmental impact of state-of-the-art models.

From the perspective of data cost, IM can be integrated into the training of LLMs(For the reason that RLHF (MacGlashan et al., 2017) is a part of LLMs). Given the high dimensionality and complex distribution of data in training LLMs, compared to the same data level, the introduction of IM may significantly improve training efficiency. This, in turn, leads to higher-quality answers in LLM-based Question and Answer (Q&A) systems, rather than relying solely on the continuous expansion of data volume. IM is therefore accessible to a broad range of researchers (even those without access to large datasets) and leaves a smaller carbon footprint. Technically, IM can be applied in multi-agent algorithms to facilitate the propagation of information across agents, thereby enhancing the coordination of multi-agent systems. For instance, in the field of autonomous driving, vehicles can simulate the driving behavior of other vehicles to better coordinate traffic flow, reduce congestion, and accidents.

Furthermore, IM is similar to an attention mechanism and can be embedded into multiple layers of DNN. It leverages the inter-sample relationships to enhance the utilization efficiency of samples. It enables the network to learn the correlation information among samples in the training set, which remains consistent in the test set, thereby improving performance and generalization on the test set.

It's fair to acknowledge that, despite the findings in this paper, we are still a long way from making Deep RL practical for solving complex real-world robotics problems. However, we believe that this work represents progress toward that objective.

If the system's administrator sets an unfair reward for vulnerable groups (or possibly even unintentionally harming themselves), the stronger the RL capabilities, the more unfavorable the results become. That's why, alongside improving algorithms for achieving cutting-edge performance, it's essential to also focus on additional research regarding fairness (Li & Liu, 2022; Xing et al., 2021).

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
