# OpenReview forum: "Imagination Mechanism: Mesh Information Propagation for Enhancing Data Efficiency in Reinforcement Learning"
_ICLR.cc/2024/Conference — Submitted to ICLR 2024_

### Official Review · Reviewer_drgh · 2023-10-24

**Soundness:** 3 good
**Presentation:** 2 fair
**Contribution:** 2 fair
**Rating:** 3
**Confidence:** 3

**Summary:**

The paper presents a novel mesh information propagation mechanism, termed the ’Imagination Mechanism (IM)’, designed to significantly enhance the data efficiency of RL algorithms. IM works as a plug-and-play module that can run seamlessly and smoothly integrated into other widely adopted RL algorithms. The authors demonstrate the improvement of SAC, PPO, DDPG and DQN under IM addition on various continuous RL tasks.

**Strengths:**

The idea of IM for supporting the mutual information transmission across episodes is interesting and (to the best of my knowledge) novel. The method focuses on minibatch in the replay buffer and has only a limited relationship to the critic network, giving it a plug-and-play advantage.

**Weaknesses:**

1.The paper's central assumption feels reasonable, and the experiment seems to confirm it. But there is no theoretical proof. The paper is not sufficient in disassembling and verifying the advantages of the IM method, so it is recommended to focus on discussing in the ablation study of the experiment.
2.The paper emphasizes the extensive enhancement of the IM for most existing RL algorithms, but the most recent description of the mainstream RL algorithms in related work is SAC in 2018. It's a bit of an overstatement and lacks a comparison of the latest work, especially non-pixel-based approaches to solving the data efficiency.
3.The IM introduced in this paper will bring additional computing and storage overload, and whether there can be related ablation study to eliminate this part of interference.
4.This paper emphasizes the advantages of solving high-dimensional state spaces and large-scale problems, but the continuous Mujoco task and discrete Gym task selected in the experimental part cannot represent the above problem scenarios. Meanwhile, the last paragraph on page 6 emphasizes model-free RL, which is inconsistent with the scope of limitations described in the full paper.
5.The experiments in this paper lack the description of the structure of the method-dependent neural network and the hyperparameters setting.

**Questions:**

1.The paper highlights in the last paragraph on page 3 that earlier work could lead to 'catastrophic learning failures'. The origin of this argument is not clear, and this paper does not prove the advantage of IM to solve this problem through experiments.
2.The description of algorithm 1 is not clear and standardized, and it lacks relevant neural network. Intuitively, it seems that there is no update process of IM introduced network.
3.In the last paragraph of the paper on page 8, the description of the experimental results of ‘Pendulum’ task, directly attributed to the limited state space, such a description is inadequate in the absence of similar experiments or evidence.

---

### Official Review · Reviewer_Cj8r · 2023-10-29

**Soundness:** 2 fair
**Presentation:** 1 poor
**Contribution:** 2 fair
**Rating:** 3
**Confidence:** 4

**Summary:**

This paper proposes a method for enhancing the sample efficiency of deep reinforcement learning methods.
The method can be thought of as data augmentation for the critic.
For two states from two different episodes, data from one state is reused to generate a target for the other state.
This is in contrast to traditional actor critic methods where values of states are only updated using data from the same episodes.
The propagate information from one episode to another episode in the update of the critic, the authors propose to measure the similiarites between two states using a similarity calculation network, which outputs a vector of K scalars.
Then this vector is passed into a difference inference network to compute the difference in the values of these two states.
This way, for state s' and action a', an additional data point can be generated by the value of state s and action a and the inferred difference:

Q(s', a') <- Q(s,a) + the inferred difference.

Experiments are performed to investigate how effective this approach is in improving the sample efficiency of various model-free DRL methods with replay buffers. A positive improvement is observed in almost all environments.

**Strengths:**

The idea seems interesting.
While I am not sure whether this idea has been explored before in the literature, I am very surprised that such a simple method can work consistently well in all the environments tested.

**Weaknesses:**

# Method

I believe important descriptions of the method are missing.
While it is clear how similiarity calculation network and difference inference network are going to be used, I didn't see any information on they are trained.

I also have questions on why would it at all.
Essentially, for the method to work, we need the difference inference network and the similiarity calculation network to generalize well across states and actions. Otherwise the artificially created targets for the critic would be misleading and potentially will harm the learning performance. Given that I can't find information on how these networks are trained, I really doubt if these networks will be able to adapt quickly and generate meaningful targets.

# Experiments

## Domain
Given that the method is so simple and the paper is making such a big claim, I would expect evaluations in more domains. For example, since you are already evaluating the method in Acrobot and Lunar Lander, why not also Cart Pole? Also, Atari games have been used as a standard benchmark for DRL methods. I would strongly suggest to at least do experiments in a few of them.

## Experimental details
There is no detail on the experiments at all.
At least, how are the hyperparameters tuned needs to be discussed to make sure of fair comparisons.
Important details about the implementation of the method and the architectures of the networks are also missing.

## Comparison to other approaches that try to improve the sample efficiency of DRL methods
Since this method focuses on improving the sample efficiency of DRL methods, I would also like to see comparisons to other approaches that try to do the same thing, to understand how effective it is.

**Questions:**

Minor:
1. wrong citation: on the bottom of page 5, He et al., 2020 should not be cited for the Adam optimizer, which is due to Kingma & Ba, 2014.

---

### Official Review · Reviewer_Wa5F · 2023-10-31

**Soundness:** 1 poor
**Presentation:** 1 poor
**Contribution:** 2 fair
**Rating:** 3
**Confidence:** 4

**Summary:**

The paper proposes a novel Imagination Mechanism (IM) to enhance data efficiency in reinforcement learning  algorithms. The authors argue that traditional RL methods rely solely on state transition information within the same episode when updating the value function, leading to low data efficiency. Inspired by human-like analogical reasoning abilities, the authors introduce IM, which enables information generated by a single sample to be effectively broadcasted to different states across episodes. The experiments on two Mujoco tasks (Halfcheetah, Ant) and three classic tasks (Pendulum, Acrobot, Lunar Lander) demonstrate that IM can improve over classic RL algorithms (SAC, PPO, DDPG, and DQN).

**Strengths:**

The paper introduces a novel approach to improve data efficiency in RL algorithms by leveraging human-like analogical reasoning abilities. The proposed IM is designed as a plug-and-play module that can be easily integrated into various existing RL algorithms.

**Weaknesses:**

1. Clarity and presentation issues. The manuscript lacks clarity in its presentation. For example, (1) Algorithm 1 not informative. It is just normal "interact and update" routines for RL with an additional call to "update Critic by IM()" with no referencing. (2) The description of the IM module would benefit from a formal presentation rather than the PyTorch-style code snippet provided in Listing 1. (3) Figure 4, which is merely a screenshot from TensorBoard based on a single seed, does not adequately represent the results. (4) The language and mathematical expressions used are not up to professional standards. For example, the term "Promotion" is  used to describe relative performance improvement in Table 1, which is not typical, and the mathematical formulas are employed in a vague and seemingly arbitrary manner in general.

2. Insufficiency in experimental validation. (1) The authors do not compare with any baselines other than standard RL algorithms. The authors should at least compare their work with [1,2,3] which has similar analytical reasoning modules. (2) There are only two tasks from Mujoco (Halfcheetah and Ant) with three classical tasks (Pendulum, Acrobot, Lunar Lander), which is not sufficient to demonstrate the general applicability of the proposed method. (3) It would be better to plot the mean and std with careful plotting rather than screenshots of the tensorboard.

3. The significance and soundnessof the proposed method are limited. There are a lot of  papers on using analogical reasoning modules to improve RL, e.g., [1,2,3], and the authors should definitely discuss and compare with them. Also, the critic $Q(s,a)$ and its value difference $d(s,a,s',a')$ are jointly learned in the IM, and it unclear why is that a meaningful objective.

Minor: attention to formatting, particularly the use of spaces, is needed. Instances like "replaybuffer" and "(4)Similarity" in the last paragraph of page 4, among others, require correction for better readability.

[1] Zhu G, Lin Z, Yang G, et al. Episodic reinforcement learning with associative memory[J]. 2020.

[2] Hu H, Ye J, Zhu G, et al. Generalizable episodic memory for deep reinforcement learning[J]. arXiv preprint arXiv:2103.06469, 2021.

[3] Li, Zhuo, et al. "Neural Episodic Control with State Abstraction." arXiv preprint arXiv:2301.11490 (2023).

**Questions:**

1. It seems that the critic and the value difference d(s,a,s',a') are jointly learned in the IM. Why is that a meaningful objective?
2. How does the method perform compared with other methods with analogical reasoning modules?
3. How does the method perform on other tasks? For example, other Mujoco tasks like Walker2d, Hopper and Humanoid or Meta-world [4] tasks.

[4] Yu, Tianhe, et al. "Meta-world: A benchmark and evaluation for multi-task and meta reinforcement learning." Conference on robot learning. PMLR, 2020.

---

### Official Review · Reviewer_1rXd · 2023-11-01

**Soundness:** 3 good
**Presentation:** 2 fair
**Contribution:** 2 fair
**Rating:** 3
**Confidence:** 4

**Summary:**

This paper proposes a new TD updation, which enables mutual information propagation across episodes. The paper extended IM to function as a plug-and-play module that can be seamlessly and fluidly integrated into other widely adopted RL methods.

**Strengths:**

1. The idea of enabling mutual information propagation across episodes is novel and interesting.

2. The empirical results are plausible, especially both in continuous and discrete environments.

**Weaknesses:**

1. The illustration figures may be misleading.

2. The writing part of the methodology is not so good.

3. There is a GitHub url in this paper, which violates the double-blind principle.

**Questions:**

1. Why directly getting information from other episodes can improve the performance, although the dynamics could be different.

2. Please compare with some other data augmentation baselines.

---

### Meta-Review · Area_Chair_d2A6 · 2023-12-02

**Metareview:**

Summary: The paper introduces "Imagination Mechanism (IM)" to enhance data efficiency in RL by enabling mutual information propagation across episodes, rather than relying solely on state transition information within the same episode. IM is presented as a versatile, plug-and-play module, compatible with existing RL algorithms like SAC, PPO, DDPG, and DQN. Empirical results demonstrate its effectiveness on simple continuous control tasks.

Strengths: The novelty of IM, its compatibility with widely-used RL algorithms, and some empirical evidence supporting its effectiveness are notable highlights.

Weaknesses: As noted by reviewers, the paper lacks clarity in its presentation, particularly in the methodology and algorithmic details. The scope of the evaluations is limited, raising questions about the generalizability of the findings. Additionally, the paper lacks a thorough comparative analysis with other methods that improve sample efficiency in RL and does not sufficiently detail the experimental setup, including network training and hyperparameter tuning. Additionally, there is an absence of a solid theoretical foundation supporting the IM's principles. The inclusion of a GitHub URL in the paper was identified as a potential violation of the double-blind review principle. Overall, these issues, coupled with potential overstatements of the IM's capabilities suggest that the paper could benefit significantly from further refinement and additional rigorous testing.

**Justification For Why Not Higher Score:**

All reviewers voted to reject the paper and I agree with their assessment.

**Justification For Why Not Lower Score:**

N/A

---

### Decision · Program_Chairs · 2024-01-16

Reject